# Genome-Wide Identification and Characterization of the *CC-NBS-LRR* Gene Family in Cucumber (*Cucumis sativus* L.)

**DOI:** 10.3390/ijms23095048

**Published:** 2022-05-02

**Authors:** Wanlu Zhang, Qi Yuan, Yiduo Wu, Jing Zhang, Jingtao Nie

**Affiliations:** 1College of Horticulture Science, Zhejiang AF University, Hangzhou 311300, China; zhangwanlu27@163.com (W.Z.); yq20gxm@163.com (Q.Y.); wuyiduo0523@163.com (Y.W.); jingzhang0411@163.com (J.Z.); 2Key Laboratory of Quality and Safety Control for Subtropical Fruit and Vegetable, Ministry of Agriculture and Rural Affairs, Hangzhou 311300, China; 3Collaborative Innovation Center for Efficient and Green Production of Agriculture in Mountainous Areas of Zhejiang Province, College of Horticulture Science, Zhejiang AF University, Hangzhou 311300, China

**Keywords:** cucumber, *CC-NBS-LRR*, bioinformatic, gene expression, miRNA

## Abstract

The NBS-LRR (NLR) gene family plays a pivotal role in regulating disease defense response in plants. Cucumber is one of the most important vegetable crops in the world, and various plant diseases, including powdery mildew (PM), cause severe losses in both cucumber productivity and quality annually. To characterize and understand the role of the *CC-NBS-LRR*(*CNL*) family of genes in disease defense response in cucumber plants, we performed bioinformatical analysis to characterize these genes systematically. We identified 33 members of the *CNL* gene family in cucumber plants, and they are distributed on each chromosome with chromosome 4 harboring the largest cluster of five different genes. The corresponding *CNL* family member varies in the number of amino acids and exons, molecular weight, theoretical isoelectric point (pI) and subcellular localization. *Cis*-acting element analysis of the *CNL* genes reveals the presence of multiple phytohormone, abiotic and biotic responsive elements in their promoters, suggesting that these genes might be responsive to plant hormones and stress. Phylogenetic and synteny analysis indicated that the CNL proteins are conserved evolutionarily in different plant species, and they can be divided into four subfamilies based on their conserved domains. MEME analysis and multiple sequence alignment showed that conserved motifs exist in the sequence of CNLs. Further DNA sequence analysis suggests that *CsCNL* genes might be subject to the regulation of different miRNAs upon PM infection. By mining available RNA-seq data followed by real-time quantitative PCR (qRT-PCR) analysis, we characterized expression patterns of the *CNL* genes, and found that those genes exhibit a temporospatial expression pattern, and their expression is also responsive to PM infection, ethylene, salicylic acid, and methyl jasmonate treatment in cucumber plants. Finally, the *CNL* genes targeted by miRNAs were predicted in cucumber plants. Our results in this study provided some basic information for further study of the functions of the *CNL* gene family in cucumber plants.

## 1. Introduction

Cucumber (*C. sativus* L.) is one of the most important vegetable crops in the world. During cucumber production, the cucumber is challenged with various serious plant diseases at different stages of development. For example, PM and downy mildew (DM) are the most common and serious diseases in cucumber plants, which lead to a significant reduction in the yield and quality of cucumbers. Therefore, one of the most desirable strategies to control these diseases is to breed resistant cultivars by cloning the resistance genes in cucumber plants. The *NLR* genes are the largest disease-resistant gene family in different crops by functioning to block pathogen invasion [1]. However, the function of these genes in cucumber plants has not been well documented. Therefore, it is of interest and importance to understand the function of the *NLR* gene family in disease response in cucumber plants.

Plants are constantly confronted by various pathogens and external adverse environment cues at different developmental stages. Therefore, plants have developed two layers of defensive mechanisms to resist the invasion of pathogens during this long-term evolutionary process [2]. PAMP-triggered immunity (PTI) is the first defensive mechanism which occurs on the plant cell membrane. It is an immunity reaction triggered by the recognition of pathogen-associated molecular patterns (PAMPs) by a variety of pattern recognition receptors (PRRs) on the plant cell membrane. However, pathogens secrete effectors in response to PTI immunity, leading some pathogens to disrupt the first line of defense of the cell membrane and enter the cell. As a stronger defense system in the cell, effector-triggered immunity (ETI) is immediately activated to prevent further invasion by pathogens. As receptors, the *NLR* genes directly or indirectly recognize the pathogen effectors [3,4]. At the same time, special responses are activated, including hypersensitivity responses (HR), cell death, and the accumulation of hydrogen peroxide.

The disease resistance gene (*R* gene) is a dominant resistant gene in plants and it can specifically detect pathogens to trigger resistance to disease [5]. The *R* gene is characterized by race specificity and higher efficiency. The NLR gene family is one of the largest *R* gene families in plants and contains nucleotide-binding sites (NBS) and leucine-rich repeat (LRR) domains [6]. They play an important role in protecting plants from various pathogens. Based on the N-terminal domain, this family can be divided into three categories: (coiled coil) CC-NBS-LRR (CNL), (toll/interleukin-1 receptor) TIR-NBS-LRR (TNL) and (resistance to powdery mildew 8, RPW8) RPW8-NBS-LRR (RNL) [7]. The sequences of this family of genes have a very important conserved domain—the NB-ARC domain (nucleotide-binding adaptor, shared by NOD-LRR proteins, APAF-1, R proteins, and the CED4 domain, also was named the NBS domain), which plays an important role in plant disease resistance and signal transduction [8,9,10,11]. The NBS domain consists of eight conserved motifs—P-LOOP (phosphate-binding loop), GLPL (Gly-Leu-Pro-Leu, also called kinase 3), RNBS-A (resistance nucleotide binding site-A), RNBS-B (resistance nucleotide binding site-B), RNBS-C (resistance nucleotide binding site-C), RNBS-D (resistance nucleotide binding site-D), kinase 2 and MHDV (Met-His-Asp-Val), and the conservation of these eight motifs is inconsistent across different plant species [12,13,14,15,16]. The C-terminal LRR structure is composed of various leucine or proline and aspartic acid amino acids that can specifically recognize proteins and participate in protein–protein interactions [17,18]. The LRR domains of NLR proteins often serve as detectors of pathogen invasion, either by directly interacting with pathogen-released effectors or by monitoring the status of effector-targeted host proteins such as RIN4 (RPM1-interacting protein 4) [19]. Upon recognition, the conformation of the NBS domain changes from an ADP-bound condensed state to an ATP-bound state exposed to the N-terminal domain, triggering a downstream hypersensitivity response that ultimately leads to apoptosis and spread of infected cells, and NBS proliferation inhibits pathogens [20].

In the NLR family, the *CNL* type is a class of important clades of plant disease resistant genes. Researches showed that *CNL* family genes can cause cell death in plants. For example, cell death was triggered in tobacco when a CNL protein (AT1G12290) was transiently expressed. However, in *Arabidopsis*, it was found that the Botrytis Susceptible1 Interactor (BOI) could regulate cell death by negatively regulating the level of AT1G12290 proteins [21,22]. The similar event of hybrid necrosis also occurs in cotton and wheat [23,24]. On the other hand, the *CNL* genes are also involved in the process of resisting invasion by different diseases in plants. For example, the protein encoded by the *CNL* genes was found to be resistant to *Verticillium wilt* in cotton and *Arabidopsis* [25,26]. In melon, *CNL* genes play a role in resistance to viral infection and aphid infection [27]. The role of the disease-resistant effects of *CNL* genes also exists in rice, wheat, and tomato [28,29,30,31,32]. In cucumber plants, two *NLR* genes, *CsRSF1* and *CsRSF2*, were reported recently to positively regulate resistance to *Sphaerotheca fuliginea* [33]. Therefore, the *CNL* genes also play a role in disease resistance in cucumber plants. However, to what extent these genes contribute disease resistance and how they function in this process largely remain elusive.

In this study, 33 *CNL* genes were analyzed in cucumber plants through the bioinformatic method. The distribution, gene cluster, and characteristics of the *CNL* family members were analyzed. The *Cis*-elements of the promoters and conserved motifs of encoded proteins of these *CNLs* were predicted. Phylogenetic and synteny analysis of the *CNL* family members in different plant species indicated that this class of genes is evolutionarily conserved. Heatmap analysis using RNA-seq data indicated that the expression of *CNL* genes was induced by various biotic and abiotic stresses. Gene expression analysis further implied that these genes exhibit a temporospatial expression pattern, and their expression is also responsive to PM infection, ethylene, salicylic acid, and methyl jasmonate treatments in cucumber plants. Moreover, the *CNL* genes targeted by miRNAs were predicted. The results provide an understanding for the further study of the function of *CNL* family of genes in cucumber plants.

## 2. Results

### 2.1. Identification and Characterization of the CNL Family of Genes in Cucumber Plants

In total, 33 *CNL* genes were identified in cucumber plants using the genome of the Chinese Long cucumber, using *Arabidopsis CNL* genes as the query sequences [34]. Their protein sequences were verified using Pfam and NCBI databases for protein functional domains. We found that the physical chromosomal locations of the 33 *CNL* genes in cucumber plants are uneven (Figure 1). The majority of *CNL* genes in cucumber plants were located on chromosome 2 with 10 *CNL* genes, followed by chromosome 2 and 3 with six *CNL* genes. Five, two, and three *CNL* genes were located on chromosomes 5, 6, and 7, respectively. However, only one *CNL* gene was located on chromosome 1. Among the 33 *CNL* genes in cucumber plants, 16 genes existed in the form of five gene clusters, approximately accounting for 48.48% of the total *CNL* genes. These five gene clusters were located at the beginning or end of chromosomes 2, 3, 4, 5, and 7. However, no gene cluster existed on chromosomes 1 and 6 (Appendix A). The largest gene cluster, consisting of five genes, was located on chromosome 4. Thus, it is likely that the probability of gene duplication on chromosome 4 was higher than that of other chromosomes in cucumber plants.

Additionally, the basic characteristics of the *CNL* genes and the encoded proteins were predicted (Table 1), including CDS length, the number of amino acids and exons, molecular weights, theoretical isoelectric point (pI), and subcellular localization. In all *CsCNL* genes, the CDS length was between 879 and 5403 bp and the number of exons ranged from one to seven. The length of the CNL proteins ranged from 293 to 1801 amino acids, and the isoelectric point was in the range from 5.57 to 9.15. In addition, the molecular weights varied significantly, ranging from 32.98491 kDa (Csa5G165310) to 207.11112 kDa (Csa2G014830). The result of subcellular localization showed that 19 (57.58%), 13 (39.39%) and 1 (3.03%) of these CNL proteins were located in the nucleus and cytoplasm, nucleus, or cytoplasm, respectively.

### 2.2. Cis-Acting Element Analysis of the Promoters of the CNL Genes in Cucumber Plants

The promoters of the 33 genes in the cucumber *CNL* family were analyzed for *cis*-acting elements. The results showed that more than 20 different *cis*-elements were found (Appendix A and Appendix A). The elements contained five kinds of plant-hormone-related elements, such as gibberellin (P-box), salicylic acid (TCA-element), abscisic acid (ABRE), methyl jasmonate (TGACG-motif, CGTCA-motif), and auxin (TGA-element). It indicated that the cucumber *CNL* family of genes may be participating in the regulation of phytohormone response. The promoter of each sequence contained one to five kinds of plant-hormone-related elements. There were 23, 15, 24, 27, and 17 genes that contained abscisic-acid-, auxin-, gibberellin-, methyl-jasmonate-, and salicylic-acid-responsive elements, respectively. It showed that one gene may have more than two of the same hormone-related elements, with the most numerous being the element response to methyl jasmonate (Appendix A). On the other hand, other important *cis*-elements also existed in these promoters, responsive to biotic and abiotic stresses, such as low temperature stress (CCGAAA), responsive elements involved in light- and drought-induced MYB binding sites (MBS); endosperm-specific negative expression (AACA_motif), meristem expression (CAT-box), mesophyll cell differentiation (HD-Zip 1), and seed-specific regulation (RY-element).

### 2.3. Analysis of the Phylogenetic Relationship and the Conserved Motifs of CNL Proteins in Cucumber Plants

To analyze the evolutionary relationships of the *CNL* gene family members, phylogenetic tree was constructed using the neighbor-joining method. The CNL proteins of cucumber, cabbage, and *Arabidopsis* were divided into 4 clades (Figure 2). There were five, and nine CNLs in CNL-A and CNL-B clades, respectively. The remaining 19 cucumber CNLs were all clustered in the CNL-C clade, while the CNL-D clade did not contain any cucumber CNLs. The amino acid sequences of the 33 cucumber CNL proteins were extracted for motif conservation analysis, and 20 motifs were screened out (Figure 3A, Appendix A). We found that most of the amino acid sequences were arranged by order of “motif13-motif9-motif7-motif16-motif14-motif20-motif11-motif2-motif1-motif10-motif3-motif5-motif8-motif6-motif19-motif4-motif17-motif15-motif18-motif12”. P-LOOP (motif7), RNBS-A (motif16), kinase-2 (motif14), RNBS-B (motif20), RNBS-C (motif11), GLPL (motif2), RNBS-D (motif1), and MHDV (motif5) were conserved regions of NBS in cucumber plants and other species (Figure 3B, Appendix A, Appendix A). However, some motifs were lost during the evolution of the CNLs. Only six sequences of CNL protein contained all the 20 motifs, and the rest of them lacked several motifs, for example, the encoded proteins of *Csa5G647580* and *Csa5G165310* only contained four conserved motifs. In addition, motif6, motif9, motif12, and motif15 were obviously either increased or deleted and shifted. Combined with a phylogenetic tree analysis, it was found that the similarity of sequence was closer, the more related motifs were contained (Figure 3A). At the C-terminus of the sequence, the motifs of the LRR family are mainly arranged in a tandem crossover manner. There were seven motifs belonging to the LRR family in the sequences of CNL proteins (motif4, motif5, motif8, motif12, motif15, motif17, and motif19). The results indicated that there are multiple LRR family motifs in one CNL protein. Moreover, a cucumber CNL protein containing all 20 motifs was aligned with homologous protein sequences from other species. We found that the conserved motifs in the NBS domain existed in different species (Figure 3B, Appendix A). The results showed that the CNL proteins were conserved in dicotyledonous and monocotyledonous plants.

### 2.4. Synteny Analysis of the CNL Genes of Cucumber

To reveal the origin and evolution of the *CNL* gene family members, we carried out a comparative syntenic analysis on *CsCNL* genes with another four plants, including melon, watermelon, tomato, and soybean (Figure 4; Appendix A). The results showed that a total of 14 *CsCNL* genes had syntenic relationship with those in melon, and 17 corresponding orthologs were identified in melon. Meanwhile, 11 *CsCNL* genes showed a syntenic relationship with those in watermelon, and 11 corresponding orthologs were identified in watermelon. Among these orthologous pairs, nine *CsCNL* genes (*Csa2G012670*, *Csa2G014830*, *Csa2G075440*, *Csa2G433370*, *Csa3G172400*, *Csa3G814390*, *Csa4G015840*, *Csa4G638480*, and *Csa5G165310*) had their corresponding orthologs both in melon and watermelon, suggesting that these genes might play an important role in the evolution of the *CNL* genes. Moreover, three and eight pairs of orthologous *CNLs* were identified between cucumber and tomato, and cucumber and soybean, respectively (Figure 4; Appendix A). The results showed that *Csa6G490170* was related to at least two pairs of homologs in tomato and soybean. The above results showed that the *CNL* genes showed stronger homology among cucumber, melon, and watermelon, than that among cucumber and two other species, which corresponded to the fact that they belonged to *Cucurbitaceae* crops.

### 2.5. Expression Analysis of the CNL Family Genes in Cucumber Plants

Using the published RNA-seq data on the CuGenDB website, the expression level of the 33 *CNL* genes in different cucumber organs or tissues (root, stem, leaf, male flower, female flower, ovary, and tendril) was summarized (Figure 5, Appendix A). The result showed that *Csa7G420890* was highly expressed in roots, leaves, and female flowers. However, some genes, such as *Csa2G012670* and *Csa2G014830*, were only highly expressed in the roots. The expression levels of some genes in all seven tissues were very low, such as *Csa5G266890*, *Csa5G647580*, and *Csa6G375730*.

According to the analysis of the *cis*-acting elements of the promoter, the expression level of the *CNL* family of genes might be affected by biotic and abiotic stresses. To verify whether *CNL* genes are involved in responding to stress, the expression level of the *CNL* genes induced by various stresses in previous studies was summarized using heatmaps (Figure 6). The results showed that the expression level of some genes had no obvious difference under stress by DM and PM, such as *Csa2G433370*, *Csa5G266890*, *Csa5G647580*, *Csa5G647550*, and *Csa6G375730*. Some *CNL* genes were up-regulated after infection with DM in cucumber plants, such as *Csa5G647590,* and *Csa3G172400* (Figure 6A, Appendix A). We found that the expression level of all genes in resistant materials (SSL508-28) was higher than that in susceptible materials (D8) at the same time point after PM inoculation (Figure 6B, Appendix A). For example, the expression level of the *Csa4G016430* gene in the SSL508-28 inbred line was higher than that in the D8 inbred line at 0 and 24 hpi (hours post-inoculation). In addition, *CNL* genes were also involved in abiotic stress responses. The expression of cucumber *CNL* genes is up-regulated under salt treatments, especially the expression of *Csa2G012670*, which is significantly increased (Figure 6C, Appendix A). Most *CNL* genes are generally up-regulated when induced by low temperatures, while the expression levels of minority genes were unchanged or down-regulated (Figure 6D, Appendix A). Taken together, these results suggested that *CNL* genes might be involved in responding to biotic and abiotic stresses.

### 2.6. Analysis of the Expression of CNL Genes by qRT-PCR

Based on the above results, six typical *CNL* genes were selected to verify their tissue-specific expression patterns by qRT-PCR (Figure 7). The results showed that these genes were all highly expressed in roots, which was consistent with the above results. *Csa3G684170* had relatively high expression in other tissues, except for the stem, cotyledons, and hypocotyl. Compared with the gene expression level in cotyledons, *Csa4G016460* had higher expression levels in other tissues. The expression level of *Csa4G638480* was lower in leaves and fruits, but higher in other tissues. The expression level of *Csa7G420890* was relatively higher in roots, stems, hypocotyls, and female flowers, and the lowest in fruits. *Csa2G014830* and *Csa2G435460* had the highest expression in roots and low expression in other tissues. The results indicated that *CNL* genes might play different roles in the growth and development of cucumber.

On the other hand, the induced expression patterns of these genes responding to PM was performed using a pair of near-isogenic lines, S1003 (resistant inbred line) and NIL(*Pm5.1*) (susceptible inbred line) (Figure 8) [36]. The expression of *Csa4G016460*, *Csa3G172400,* and *Csa4G638480* in S1003 were significantly lower than that in NIL(*Pm5.1*) at 0 hpi. Except for the above genes, the expression of the other three genes (*Csa2G014830*, *Csa2G435460* and *Csa3G684170*) had no difference at 0 hpi between S1003 and NIL(*Pm5.1*). The expression of *Csa4G016460* was continuously decreased after inoculation in two lines. The expression level of *Csa4G016460* in S1003 was significantly higher than that in NIL(*Pm5.1*) at 12, and 24 hpi, but the expression of *Csa4G016460* in NIL(*Pm5.1*) at 48 hpi was higher. *Csa2G014830*, *Csa2G435460*, *Csa3G684170, Csa3G172400*, and *Csa4G638480* had similar expression patterns post-inoculation. Their expression level reached the highest at 12 hpi and then decreased in S1003 and NIL(*Pm5.1*). It also found that the expression of *Csa2G014830* was significantly lower at 12 hpi, and 24 hpi in NIL(*Pm5.1*) than that in S1003, and there was no difference at 48 hpi between the two lines. Except for 0 and 24 hpi, there was a significant difference in the expression level of *Csa2G435460* between S1003 and NIL(*Pm5.1*). The expression of *Csa2G435460* in S1003 had a significantly higher level than that in NIL(*Pm5.1*) at 12 hpi, but had an opposite expression pattern at 24 hpi. *Csa3G684170* had a similar expression pattern in two materials while the expression was significantly higher in NIL(*Pm5.1*) than S1003 at 24 hpi. There were no significant difference in the two lines at 48 hpi. The expression patterns of *Csa3G172400* and *Csa4G638480* were similar between S1003 and NIL(*Pm5.1*). The expression levels of *Csa3G172400* and *Csa4G638480* were significantly lower at 0 hpi at S1003. In addition, the expression levels of *Csa3G172400* and *Csa4G638480* were significantly lower at 12 hpi at NIL(*Pm5.1*), while they were significantly higher at 24 hpi and 48 hpi at NIL(*Pm5.1*). Five *CNL* genes reached highest expression level at 12 hpi, which showed that they might be corelated with the PM resistance conferred by *Csmlo1*. These results showed that the *CNL* genes might be involved in PM resistance.

### 2.7. Expression Patterns of the CNL Genes Responding to Hormones in Cucumber Plants

The *cis*-elements of the predicted promoters of the *CNL* genes showed that the promoters of most *CNLs* contained 1–5 *cis*-elements responding to phytohormones. To further analyze the effect of phytohormones on the expression of *CNL* genes, the cucumber was treated with different exogenous hormones, which related to stress, and the expression patterns of seven *CNL* genes were selected to be monitored within a short period of time after hormone application.

The results showed that the expression level of seven *CNL* genes was changed, ranging from 0 h to 24 h after ethylene treatment (Figure 9). Specifically, it was found that expression of *Csa3G684170*, *Csa2G014830*, and *Csa3G172400* was firstly decreased and then increased, and the expression reached the highest level at 12 h. The expression patterns of *Csa2G420890* and *Csa4G016460* were similar to the previous three genes, but the expression reached the highest level at 24 h after ethylene treatment. The gene expression levels of *Csa4G638480* and *Csa2G435460* were increased firstly and then decreased, and the expression reached the highest level at 12 h. It was speculated that, within 24 h of exogenous ethylene application, the *CNL* genes showed various expression patterns responding to ethylene, which might balance the relationship between development and the response to the environment. 

Salicylic acid (SA) had been reported to respond to various disease resistance mechanisms in different plants [34,37,38]. Therefore, salicylic acid treatment was carried out on cucumber seedlings to further study the effect of salicylic acid on cucumber *CNL* gene expression by qRT-PCR (Figure 10). The expression pattern of the seven *CNL* genes showed a similar tendency responding to salicylic acid. The expression level of these genes increased continuously and then decreased back to normal. The difference in the pattern was that expression level of *Csa3G684170*, *Csa2G014830*, *Csa3G172400,* and *Csa2G420890* reached the highest at 12 h, while the expression level of the other three genes reached the highest at 6 h after salicylic acid treatment. These results indicated that the expression of the *CNL* genes was induced by salicylic acid treatment in the early stages of treatment, especially at 6 h and 12 h.

Finally, aiming to analyze the effect of methyl jasmonate on the expression of *CNL* genes, the expression pattern of the *CNL* genes was monitored at 0–24 h after methyl jasmonate treatment, by qRT-PCR (Figure 11). We found that the expression levels of all genes kept relatively low at 0 h, 6 h, and 12 h after treatment, and the expression levels were largely increased at 24 h after treatment. The above results indicated that the expression of *CNL* genes was induced by different exogenous hormones.

### 2.8. Prediction Analysis of the Binding Site of CNL Genes Targeted by miRNA in Cucumber Plants

In order to determine whether the cucumber *CNL* genes were regulated by microRNA through a targeted binding site, the published miRNAs sequences were used to predict the binding site of *CNL* genes targeted by miRNA on the psRNATarget website [39,40]. The miRNA sequencing and prediction analysis has been performed in a previous study. The mature miRNA sequence was used for analysis of the binding site of the *CNL* genes. It turned out that 30 genes might be regulated by 33 kinds of microRNA (Appendix A and Figure 12). Among them, 17 *CNL* genes might be targeted by miRNA482, accounting for 17% of the *CNL* genes in cucumber plants. This is followed by the number of *CNL* genes targeted by miRNA2118, miRNA396, and miRNA156, accounting for 9%, 9%, and 8% of all CNL genes, respectively. Six, five, and five *CNL* genes were targeted by miRNA157, miRNA172, and miRNA395, respectively. In addition, it was shown that most of the interactions between the targeted *CNLs* and various miRNAs were at the transcriptional level, and only a few of them were at the translational level. We also found that several miRNAs could target one *CNL* gene, such as *Csa4G638480*, the homologous gene of *AtADS1* in *Arabidopsis*, targeted by miRNA396, miRNA164, and miRNA390 (Appendix A). Meanwhile, multiple *CNL* genes might be targeted by one miRNA, for example, the miRNA482 could target 17 *CNL* genes (Appendix A).

## 3. Discussion

In this study, a total of 33 *CNL* genes were obtained from cucumber plants by the bioinformatics method (Figure 1 and Table 1), accounting for 0.135% of the whole gene numbers in cucumber plants. There were 455, 111, and 89 *CNL* genes in rice, tomato, and potato plants, respectively [41,42,43]. In addition, they accounted for 0.994%, 0.310%, and 0.153% of each gene numbers, which was larger than the proportion of *CNL* genes in cucumber plants. There were 69 *CNL* genes found in maize, with 0.089% proportion of the whole genes, which was lower than the proportion of *CNL* genes in cucumber plants [44]. In cucumber plants, 48.48% of the *CNL* genes were presented on the chromosome in the form of gene clusters (Appendix A). This evolution phenomenon might have a positive effect on defending from various pathogens in nature. Evolutionary events, such as unequal crossover, insertion/deletion, and gene conversion, occurring in the *NLR* genes provided the possibility for increasing the rate of mutation and the formation of denser gene clusters on chromosomes, which has been found in lettuce and radish plants [45,46,47]. Based on the prediction of gene structure and protein physicochemical properties, we found that *CsCNL* genes had similar characteristics to the *CNL* genes in Chinese cabbage, cabbage and kiwifruit [1,13,48]. We speculated that the function of the CNL proteins may be affected by their characteristics.

*Cis*-elements play an important role in various life processes, such as plant growth and development, stress response, hormones response, and signal transduction. The promoters of *CNL* genes were analyzed for *cis*-acting elements (Appendix A, Appendix A), and we found that many important *cis*-elements existed in the promoters, such as various plant-hormone-related elements, light-related elements. Among them, 16 gene members contained an element named “W-box”, which is the DNA binding site of the WRKY protein induced by salicylic acid [38]. Research suggested that it could positively regulate *RPP8*, a DM resistance gene, in *Arabidopsis* [49]. The results suggested that the *CNL* genes might be implicated in the defense against certain fungal diseases, and the *CNL* genes might also play a role in defense against various diseases in cucumber plants. To verify the roles of these hormonal elements in disease resistance, we monitored the expression patterns of the *CNL* genes under different hormone treatments by qRT-PCR (Figure 9, Figure 10 and Figure 11). We speculated that the expression level responding to methyl jasmonate at 24 h was particularly high due to the multiple elements related to methyl jasmonate response. These results indicated that *CNL* genes were induced by different hormones (ethylene, salicylic acid, and methyl jasmonate).

Motif conservation analysis was performed on the CNL gene family proteins, and 20 motifs were obtained (Figure 3, Appendix A). It was found that eight conserved motifs of the NBS domain existed in most of the proteins. In addition, motif6, motif 9, motif 12, and motif 15 were obviously either increased, or deleted and shifted, which might be related to the different genetic variation generated by natural selection in the face of the various adversities in cucumber plants (Figure 3A). However, some conserved motifs of individual *CNL* genes were lost during the evolution process, for example, *Csa5G165310* lost 16 motifs in the sequence. Studies have reported that the loss rate of *NLR* genes in cucumber plants was higher than that of other plants. Because the copy number of cucumber *NBS-LRR* genes was affected by gene loss, this resulted in further diversity of the resistance conferred by them [50].

Synteny analysis showed that a higher synteny was maintained among cucurbit crops (Figure 4, Appendix A). This result demonstrated the existence of more orthologous pairs between cucumber plants and other cucurbit crops, due to the closer relatedness of evolution. The results of tissue-specific expression analysis showed that the expression level of *CNL* genes was relatively high in the root, and the expression level in other tissues was varied (Figure 7). Some genes had different expression levels in different tissues. For example, *Csa2G014830* has a high expression level in the roots, while its expression level is relatively low in tendrils and other tissues (Figure 4, Figure 7). Moreover, the expression levels of different genes in the same tissue were different, such as *Csa4G016460* and *Csa7G420890*. Although the expression of some *CNL* genes in the leaves was relatively low, they were induced to higher levels when they were faced infection by PM and hormone treatment. This indicated the precise regulation of the *CNL* genes to balance the development and stress responses. Moreover, the results by qRT-PCR were slightly different from the previous studies, which might be caused by using a different cucumber inbred line. Heatmap analysis using published data found that the expression of cucumber *CNL* genes was also increased under abiotic stresses, such as salt stress and low temperature treatment (Figure 6). The results showed that these genes might not only confer resistance to pathogens, but also play a role in abiotic stresses. Inoculation experiments of PM pathogens were performed using one pair of NIL in cucumber plants, and gene expression at different time points after inoculation was detected by qRT-PCR (Figure 8). The expression was very low at 0 hpi, but the gene expression level of the *CNL* genes (such as *Csa3G172400*) increased rapidly after inoculation and reached the peak at 12 hpi. Previous studies have confirmed the existence of a recessive disease resistance gene in cucumber plants, *Csmlo1*, which could confer durable resistance to PM in cucumber plants [51]. The expression of the *CsMLO1* gene reached the highest level at 12 hpi of PM, which indicated that the *CsMLO1* gene played a key role in responding the attack of PM at this period post-inoculation of PM on cucumber plants. According to the induced expression results of the above *CNL* genes, we found that five genes have a similar expression pattern to *CsMLO1*. These results indicated that the functions of these genes might correlate with the *Csmlo1* resistance pathway in the process of resisting the invasion of PM. The expression of these *CNL* genes was strictly controlled at a low level when there were no pathogens invading [51]. This was to prevent damage to plants caused by constitutive activation and spontaneous reaction, such as hypersensitivity reactions and reactive oxygen species (ROS) production [52]. On the other hand, the *CNL* family of genes were known as a class of dominant disease-resistance *R* genes, and they play a role in the plant’s defense mechanism (PTI and ETI) [4,53]. The above results indicated that the *CNL* genes might be involved in disease resistance pathways controlled by dominant genes, and might also be coupled with the *Csmlo1* pathway that plays a role in PM resistance, or the other diseases resistance pathways. However, these assumptions still need further experiments to be carried out for confirmation. Therefore, the study of *CNL* genes is of great significance for cultivating durable and stable disease-resistant cucumber varieties.

The miRNAs are a class of small non-coding RNAs about 20-24nt in size, they mainly function to regulate gene expression at the post-transcription level in both plants and animals [54]. They play important roles in various life stages of plants, for example, the conserved miR156–miR172 were reported to regulate plant vegetative phase changes [55]. The miRNAs have been shown to be responsive to various biotic and abiotic stresses [56,57,58,59,60], including pathogen infection. Furthermore, miR396 negatively regulates rice blast resistance by inhibiting multiple *OsGRFs* genes [61], and miR482 is an ancient and conserved miRNA family of 22nt in length, and it plays an important role in the development and the process of resistance to diseases and stresses [62]. Studies have reported that miRNA482 could participate in the defense against pathogens by regulating their targeted genes, such as miR482 targets of the mRNA of *CNL* genes in cotton [39]. This study indicated that only 12% of *NLR* genes were targeted by the members of the miR482 family, and miRNA482 was inhibited when its targeted *NLRs* were induced to defend against pathogen invasion [39]. In the presence of pathogens, miR482 was silenced and the expression of the targeted genes *NLR* were increased, resulting in plant resistance. Specifically, miRNA482 cleaves the P-LOOP motif of the target *NLR* genes and produces a large amount of phasiRNA, which can enhance the silencing effect of miR482 on *NLR* genes. Thus, miR482 and its target genes *NLR*, as well as the generated phasiRNA, are involved in the process of plant immunity [37]. Moreover, the known miRNAs sequences were used to predict the possible target sites on cucumber *CNL* genes. It was found that one *CNL* gene could be targeted by different miRNAs, and the same miRNA could also target multiple *CNL* genes. Among them, miRNAs (such as miRNA156 and miRNA172) might regulate *CNL* genes and play an important role in growth and development. In addition, miRNA156 has been proven to function in the development of the age pathway and it is involved in responding to biotic stresses [55,63]. Importantly, miRNA482 was predicted to target 17 *CNL* genes in cucumber plants (Appendix A). Previous studies have shown that miRNA482 could negatively regulate the *CNL* gene and play an important role in plant disease resistance [64]. These results indicated that miRNA482 and *CNL* genes might also be correlated in disease resistance in cucumber plants, and the *CNL* genes might be regulated by miRNA to balance the development and defense against pathogens. Therefore, the present study provided a direction for further research on the mechanism of action between miRNAs and *CNL* genes in cucumber plants. According to the previous studies, miRNA482 could cleave the target genes, for example, the *CNL* genes, and generate a large number of phasiRNAs, which were then involved in the response to pathogenic invasion in the plant’s ETI immune system. Therefore, we speculated that these *CNL* genes might be regulated by miRNAs, and they could cooperate with each other to play a role in disease resistance in cucumber plants.

## 4. Materials and Methods

### 4.1. Identification of Cucumber CNL Family Genes

Firstly, blast homologous sequences of *CNL* in cucumber plants were searched on the CuGenDB website (http://cucurbitgenomics.org/, accessed on 14 March 2022) by using the *Arabidopsis CNL* family of genes as the query sequences [65]. Secondly, the NB-ARC domain was identified by PF00931 on the Pfam website (http://pfam.xfam.org/, accessed on 22 February 2022), then homologous protein sequences from *Arabidopsis* were scanned for “hmmsearch”. Then, the candidates and conserved domains of CNLs were confirmed by a NCBI CD search (https://www.ncbi.nlm.nih.gov/Structure/bwrpsb/bwrpsb.cgi, accessed on 8 March 2022). In addition, the coiled-coils structure was confirmed on the Paircoil2 website (http://cb.csail.mit.edu/cb/paircoil2/paircoil2.html, accessed on 25 February 2022), and the P-value parameter was set as 0.025 [66].

### 4.2. Analysis of Gene Characteristics, Genomic Distribution, and Cis-Acting Elements in Promoters

Information concerning the *CNL* genes, including exon numbers and CDS, were retrieved from the CuGenDB website and proved using FGENESH (http://linux1.softberry.com/berry.phtml?topic=fgenesh&group=programs&subgroup=gfind, accessed on 26 April 2022). The theoretical pI and molecular weight of the identified CNLs were calculated using the ProtParam website (https://web.expasy.org/protparam/, accessed on 22 April 2022). Moreover, we predicted the subcellular localization of CNL proteins by using the CELLO website (http://cello.life.nctu.edu.tw/, accessed on 8 April 2022). The cucumber genome annotation file was downloaded from the Ensembl Plants website (http://plants.ensembl.org/index.html, accessed on 16 April 2022) for analysis. According to the gene position, TBtools was used to map the physical location of the *CNL* genes. A gene cluster was regarded as the distance between two adjacent *NLR* genes being <200 kb, with ≤8 non-*NBS-LRR* genes between the two *NLR* genes [38]. Furthermore, the promoter sequence (2 kb upstream of the gene initiation codon) of *CNL* genes was submitted to the Plant CARE website for *cis*-element analysis (http://bioinformatics.psb.ugent.be/webtools/plantcare/html/, accessed on 23 April 2022), and then visualized using TBtools.

### 4.3. Analysis of the Conserved Motif and the Synteny of the CNL

Using the MEME Suite website (https://meme-suite.org/meme/, accessed on 25 March 2022), the amino acid sequences of 33 *CNL* genes were added for motif prediction. The results were downloaded in a MEME.xml format, and then be input into TBtools software for visualization. The whole genome sequence files and gene annotation files of soybean, tomato, and maize from the Ensembl Plant website were downloaded, and the whole genome sequence files and gene annotation files of cucumber, melon, pumpkin, and watermelon from the CuGenDB website were downloaded. The collinear relationship between different species was plotted with the help of TBtools software.

### 4.4. Transcriptome Analysis of the Genes Expression of the CNLs 

The cucumber transcriptome data (PRJNA80169) was found via the cucumber genome website, and the expression levels of 33 *CNL* family genes in seven tissues, including roots, stems, leaves, male flowers, female flowers, carpels, and tendrils, were analyzed. The expression levels of these genes were plotted into a heatmap using TBtools software. Similarly, heatmaps of *CNL* gene family expression were drawn for the cucumber transcriptome data of PM treatment (PRJNA321023) [67], DM treatment (PRJNA388584), salt stress treatment (PRJNA437579), and low temperature stress treatment (PRJNA438923).

### 4.5. Tissue-Expression Analysis

Cucumber 9930 materials and qRT-PCR analysis were used for tissue-specific expression experiments. The roots, stems, leaves, cotyledons, hypocotyls, male flowers, female flowers, and fruits of cucumber 9930 were sampled. The experiment had three biological replicates and each organ from 15 cucumber plants were harvested and used as a sample in each replicate.

### 4.6. Cucumber Materials and the Treatment of PM and Hormones

Cucumber seedlings were planted in a constant temperature incubator (16 h day and eight hours night; temperature 25 °C). Before the treatment, the seedlings were cultivated in an incubator free of pathogens. Cucumber 9930 materials were used for hormone-responding experiments, and S1003 and NIL(Pm5.1) materials were used for the inoculation experiments with PM [36,68]. At the two-leaf stage, different hormones, salicylic acid (2 mmol/L), methyl jasmonate (100 μmol/L), ethylene (200 ppmol/L), and abscisic acid (100 μmol/L) were sprayed on the cucumber 9930 leaves. Each treatment had three biological replicates and the leaves of 15 plants were harvested as samples at different time points after treatments (0 h, 6 h, 12 h, and 24 h) in each replicate. At the two-leaf stage, plants were inoculated with the PM pathogen by spraying a spore suspension (1 × 10^5^ spores/mL) evenly onto the leaves. Samples were taken at 0 h, 12 h, 24 h, and 48 h after inoculation for expression analysis. The experiment had three biological replicates and the leaves of 10 plants were harvested and used as a sample at different time points in each replicate.

### 4.7. Total RNA Extraction and qRT-PCR Analysis

Total RNA from different cucumber tissues was extracted by the Triozol method, the DNA was removed, and then was reversely transcribed into cDNA using the PrimeScript first Strand cDNA Synthesis Kit (Takara, Japan). The primers for qRT-PCR were designed in Primer 5 and the *CsActin* was used as an internal control gene (Appendix A). The qRT-PCR was carried out using the SYBR Premix Ex Taq II Kit (Takara, Japan), PCR was performed on a StepOne PlusTM (ABI, America) real-time PCR machine. The qRT-PCR parameters were set as 95 °C for 10 s, 60 °C for 15 s, and 72 °C for 25 s for 45 cycles, and melting curve analysis was performed with the default settings on the instrument. The 2−ΔΔCt calculation method was used for quantitative expression analysis. Data were analyzed using normalization algorithms.

### 4.8. Prediction of Binding Sites of the CNL Genes Targeted by miRNA

Downloaded miRNA sequences were from psRNATarget (https://www.zhaolab.org/psRNATarget/, accessed on 9 April 2022) [69]. The cDNA library was chosen, with the library entitled “*Cucumis sativus* (cucumber), cds, cucumber genome sequencing project, version 2”, and the rest of the parameters were set as default. The predicted binding sites of *CNL* genes were screened out in the Excel sheet.

## 5. Conclusions

In this study, a total of 33 *CNL* genes were identified in cucumber plants. The distribution of *CNL* genes on the chromosome was uneven, some genes existed on the chromosomes in the form of gene clusters. The corresponding CNL proteins were varied in the number of amino acids and exons, molecular weight, theoretical isoelectric point (pI) and subcellular localization. Analysis of *cis*-elements revealed that the promoters of the *CNL* genes included many important elements in the growth and development of plants, such as hormone-response elements, drought-response elements, cold-response elements and damage-inducing elements. Analysis of the phylogenetic tree and conversed motifs suggested that the CNL proteins are evolutionarily conserved. Syntenic analysis indicated that more orthologous pairs existed in the *CNL* gene family among the *Cucurbitaceae* crops (cucumber, melon, and watermelon), compared to tomato and soybean plants. Heatmap analysis and tissue-specific expression analysis of the *CsCNL* genes demonstrated their diverse spatiotemporal expression patterns. Moreover, heatmap analysis also indicated that the expression of *CNL* genes were induced by various biotic and abiotic stresses. The qRT-PCR analysis showed that those genes also responded to PM infection, and treatment with ethylene, salicylic acid, and methyl jasmonate in cucumber plants. Finally, we predicted that many *CNL* genes were targeted by miRNAs, especially the 17 *CNL* genes targeted by miRNA482. Our results provided the basis for further study of the function of *CNL* genes in cucumber plants.

## Figures and Tables

**Figure 1 ijms-23-05048-f001:**
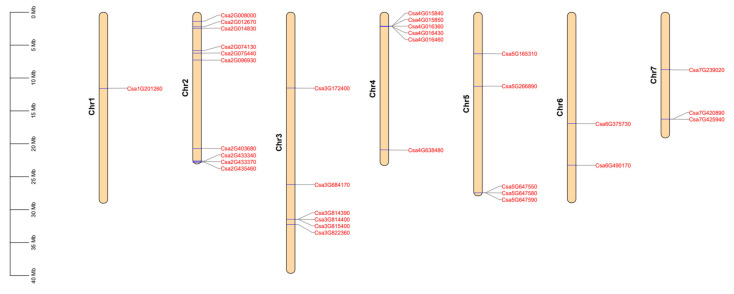
The distribution of the *CNL* genes located on the chromosomes in cucumber plants. The genetic distance of seven chromosomes were represented by the scale in megabases (Mb) on the left. The *CNL* genes are displayed using nomenclature for genome version 2 of Chinese Long cucumber. Blue lines represent the location of the gene on each chromosome.

**Figure 2 ijms-23-05048-f002:**
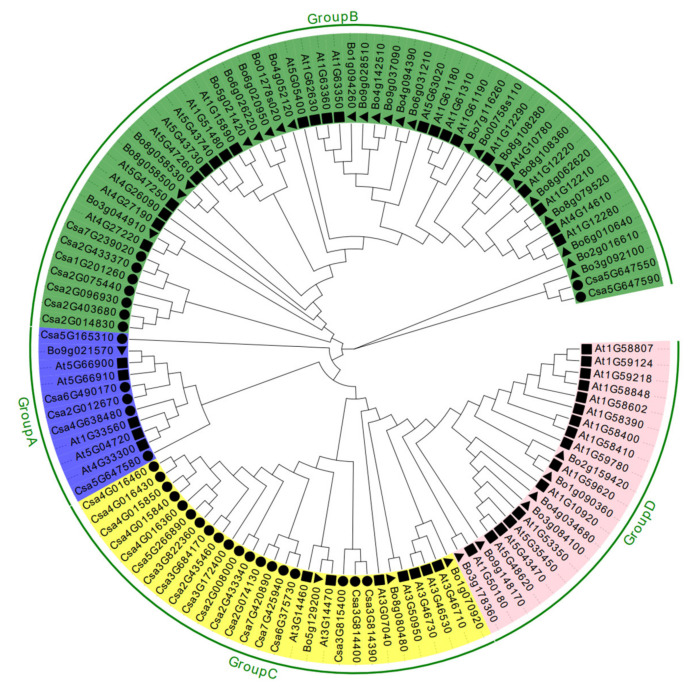
Phylogenetic relationships of CNL proteins among cucumber, cabbage and *Arabidopsis*. The unrooted phylogenetic tree was constructed by MEGA7.0 by neighbor-joining method with 1000 bootstrap replicates. The CNLs are divided into four major subfamilies. Different subfamily was indicated with different colors. The black triangles represented cabbage, black squares represented *Arabidopsis*, and black circles represented cucumber.

**Figure 3 ijms-23-05048-f003:**
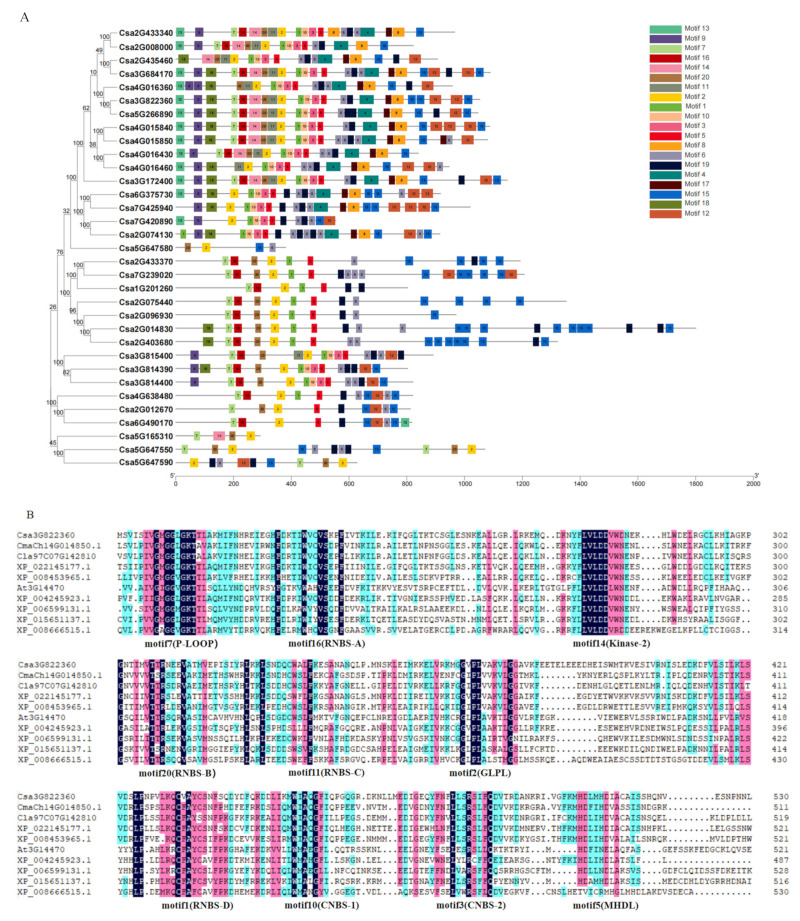
Analysis of the conserved motifs of CNL proteins. (**A**) MEME analysis of the CNL proteins in cucumber plants. (**B**) Alignment of the conserved NBS domain of CNL proteins between cucumber and other species. Black color: the homolog level was 100%; Pink color: the homolog level was greater or equal to 75%; Blue color: the homolog level was greater or equal to 50%. Maize, *XP_008666515.1*; bitter melon, *XP_022145177.1*; melon, *XP_008453965.1*; tomato, *XP_004245923.1*; soybean, *XP_006599131.1*; pumpkin, *CmaCh14G014850.1*; *Arabidopsis*, *At1G10920*; watermelon, *Cla97C07G142810.1*; and rice, *XP_015651137.1*.

**Figure 4 ijms-23-05048-f004:**
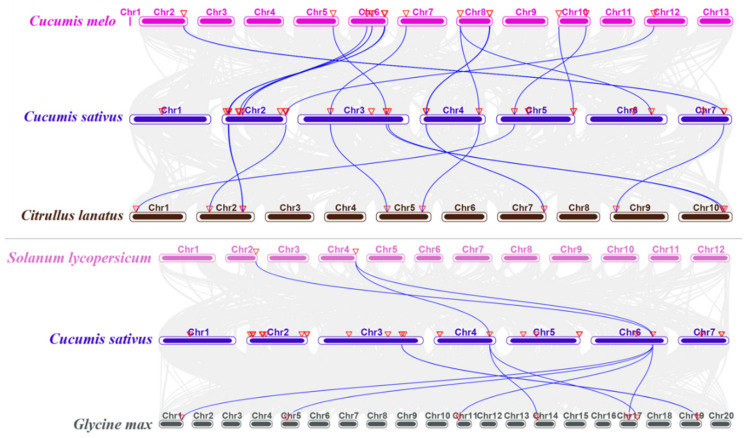
Synteny of the *CNL* genes among cucumber and other species. Gray lines in the background represent collinear blocks in cucumber plants and other genomes. The collinear gene pairs with *CNL* genes between different species were highlighted by the blue lines. Red inverted triangle indicated the locations of the *CNL* genes. Cucumber (*C. sativus* L.), melon (*Cucumis melo* L.), watermelon (*Citrullus lanatus* L.), tomato (*Solanum lycopersicum* L.), and soybean (*Glycine max* L.).

**Figure 5 ijms-23-05048-f005:**
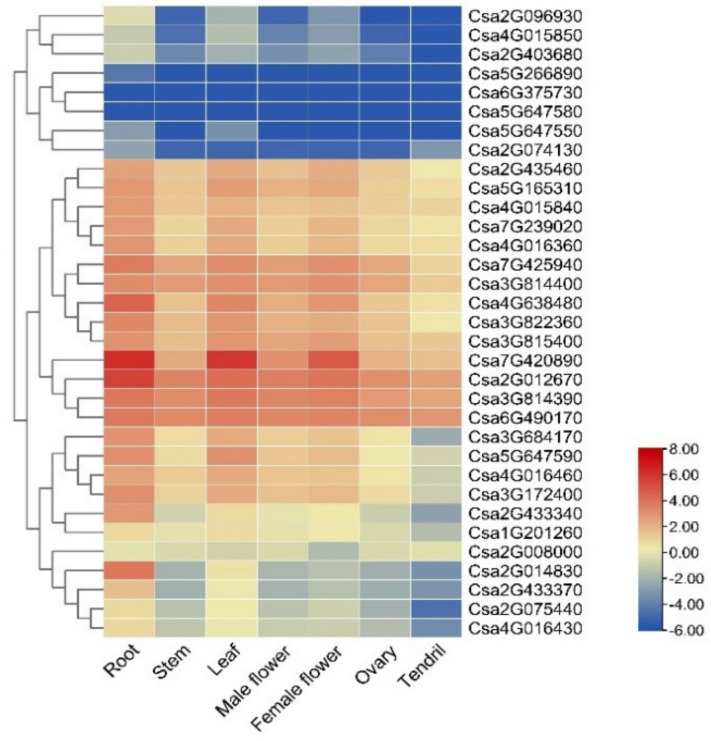
Tissue-specific expression of the *CNL* genes in cucumber plants. The transcriptional levels of *CsCNL* genes in seven tissues of cucumber 9930 were investigated based on public transcriptome data (PRJNA80169) [35]. The genome-wide expression of *CsCNL* genes were shown on a heatmap using a log_2_RPKM value, and −6.00 to 8.00 was artificially set with the color scale limits according to the normalized value. The color scale showed increasing expression levels from blue to red.

**Figure 6 ijms-23-05048-f006:**
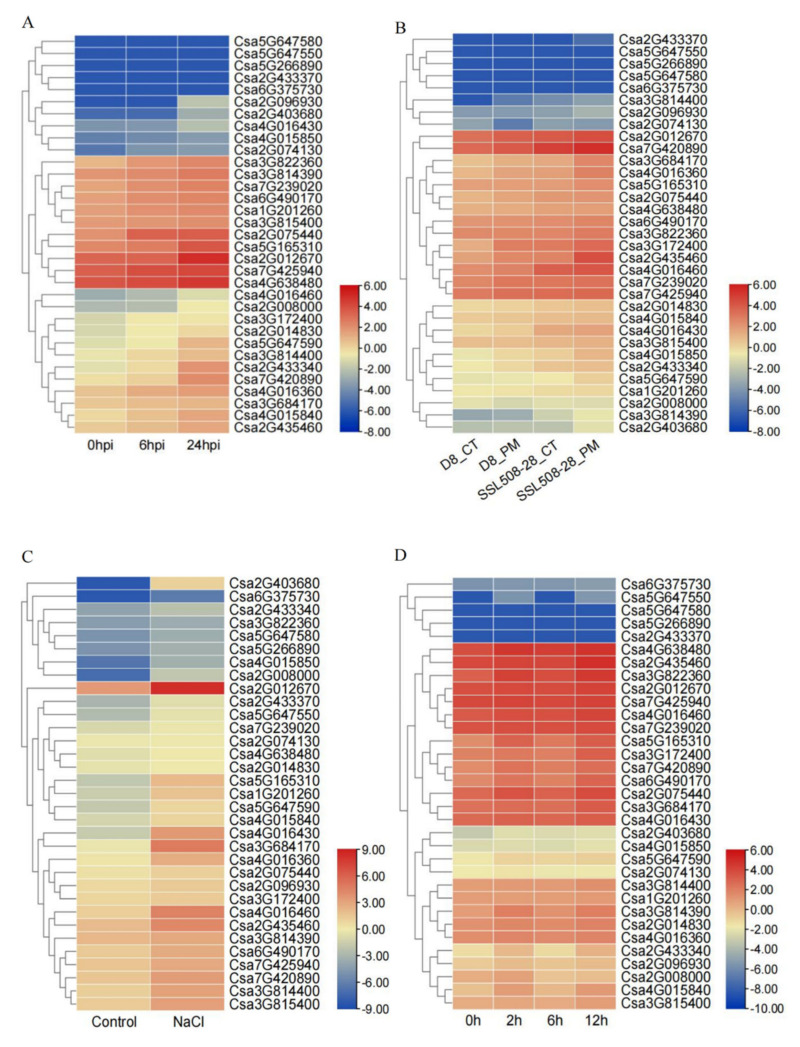
The expression patterns of the *CNL* genes under biotic and abiotic stresses in cucumber plants. (**A**) The expression of cucumber *CNL* genes in response to DM, post-inoculation at 0 h, 6 h, and 24 h. (**B**) The expression of cucumber *CNL* genes in response to PM, post-inoculation at 0 h, and 48 h. D8_CT, D8 line control; D8_PM, D8 line, 48 h post-inoculation with PM; SSL508-28_CT, SSL508-28 line as control; SSL508-28_PM, SSL508-28 line, 48 h post-inoculation with PM. (**C**) and (**D**) The expression of cucumber *CNL* genes in response to salt stress and chilling stresses. Hpi, hours post-inoculation. The genome-wide expression of *CsCNL* genes were shown on a heatmap using log_2_RPKM value, and −8.00 to 6.00, −9.00 to 9.00, −10.00 to 6.00 were artificially set with the color scale limits according to the normalized value. The color scale was shown increasing expression levels from blue to red.

**Figure 7 ijms-23-05048-f007:**
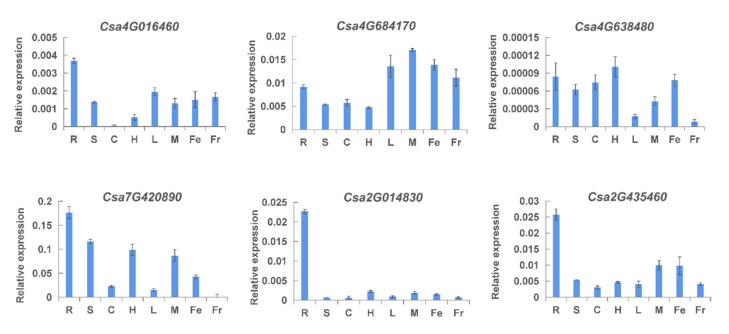
Tissue-specific expression analysis of the *CNL* genes in cucumber plants by qRT-PCR. R: root; S: stem; C: cotyledon; H: hypocotyl; L: leaf; M: male flower; Fe: female flower; and Fr: fruit. The vertical axis is relative to the expression level and *x*-axis represents different tissues. Values are mean ± SE (n = 3).

**Figure 8 ijms-23-05048-f008:**
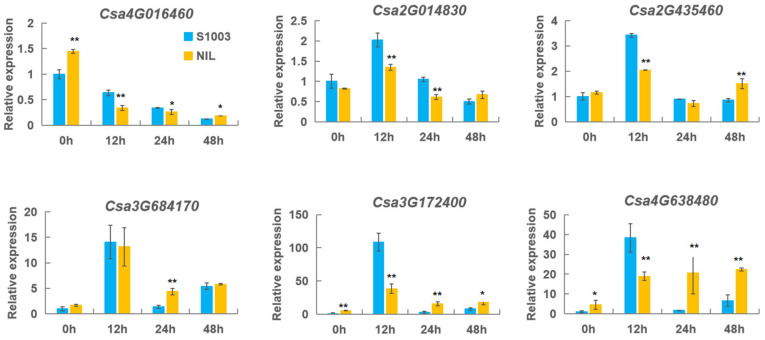
The expression patterns of the *CNL* genes responding to PM in cucumber plants. The expression levels of the *CNL* genes were detected in S1003 and NIL(*Pm5.1*) for 0 h, 12 h, 24 h, and 48 h after inoculation. S1003, resistance cucumber inbred line; NIL, and susceptible cucumber inbred line NIL(*Pm5.1*). The expression level of transcript at 0 h was set to a value of ‘1′. Values are mean ± SE (n = 3) (* and ** indicate significant differences between S1003 and NIL(*Pm5.1*) at *p* = 0.01 and 0.05, respectively).

**Figure 9 ijms-23-05048-f009:**
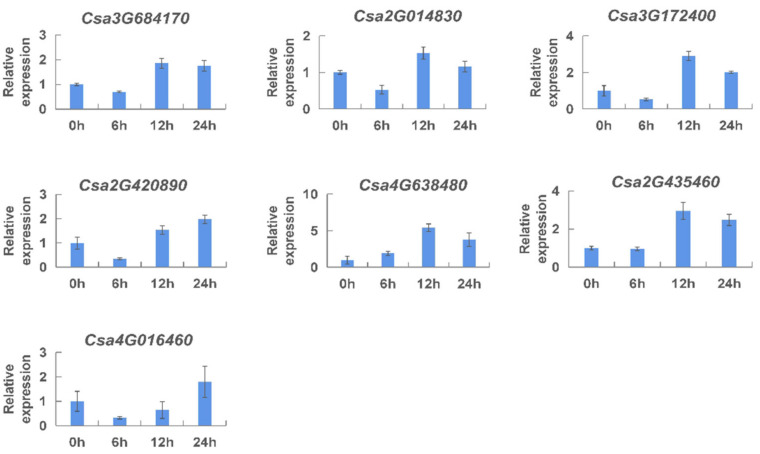
Expression analysis of the *CNL* genes responding to ethylene treatment in cucumber plants. The vertical axis is relative expression level and 0, 6, 12, 24 h on the *x*-axis indicate the treatment time. The expression level of transcript at 0 h was set to a value of ‘1′. Values are mean ± SE (n = 3).

**Figure 10 ijms-23-05048-f010:**
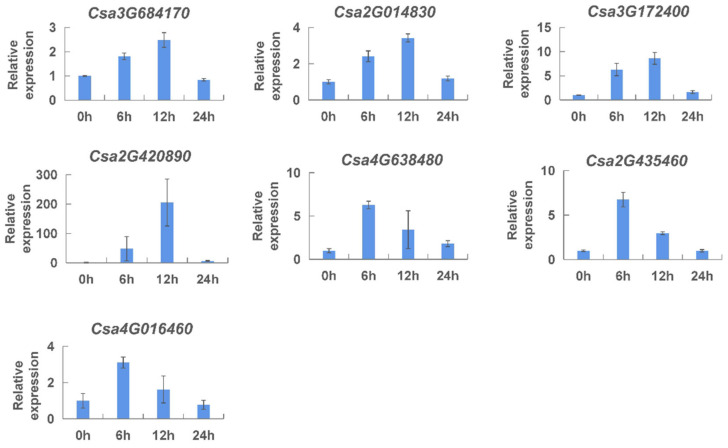
Expression analysis of the *CNL* genes responding to salicylic acid treatment in cucumber plants. The vertical axis is relative expression level and 0, 6, 12, 24 h on the *x*-axis indicate the treatment time. The expression level of transcript at 0 h was set to a value of ‘1′. Values are mean ± SE (n = 3).

**Figure 11 ijms-23-05048-f011:**
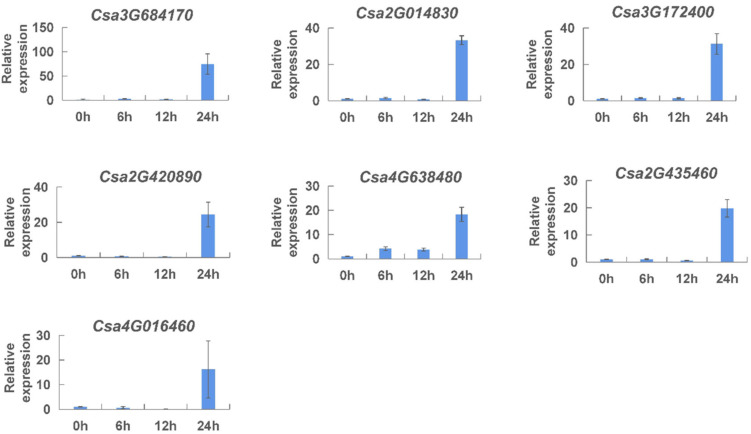
Expression analysis of the *CNL* genes responding to methyl jasmonate in cucumber plants. The vertical axis is relative expression level and 0, 6, 12, 24 h on the *x*-axis indicate the treatment time. The expression level of transcript at 0 h was set to a value of ‘1′. Values are mean ± SE (n = 3).

**Figure 12 ijms-23-05048-f012:**
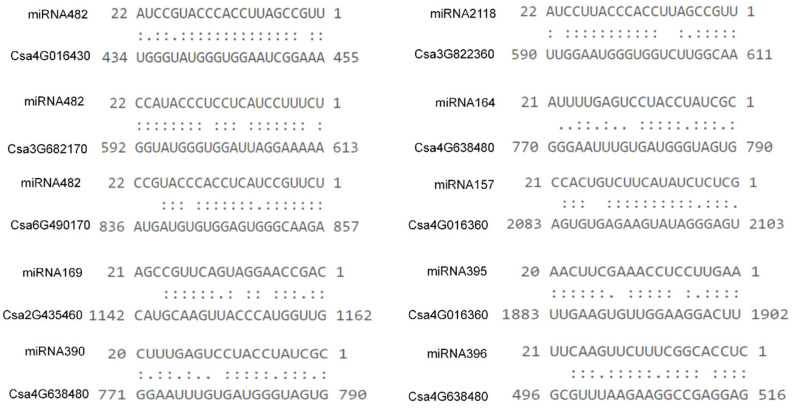
The predicted binding site of the *CNL* genes targeted by miRNAs in cucumber plants. Two dots indicated paired successfully between bases, and one dot indicated that there is also a pairing between U and G in the secondary structure. Blank space indicated that two bases failed to be paired.

**Table 1 ijms-23-05048-t001:** Characteristics of the *CNL* family of genes and the corresponding proteins in cucumber plants.

Gene ID	Length of CDS (bp)	Number of Amino Acids	Number of Exons	Molecular Weight (kDa)	Theoretical pI	Subcellular Localization
*Csa1G201260* *Csa2G008000*	2409 2469	803 823	7 5	90.94932 95.32828	5.72 6.80	N/Cyto Cyto/N
*Csa2G012670*	2439	813	5	92.46041	7.93	N
*Csa2G014830*	5403	1801	7	207.11112	5.57	N
*Csa2G074130*	2745	915	2	105.95585	8.29	N
*Csa2G075440*	4056	1352	4	152.92906	6.35	N/Cyto
*Csa2G096930*	2913	971	1	110.91379	6.20	N/Cyto
*Csa2G403680*	3966	1322	4	151.81125	6.54	N
*Csa2G433340*	2898	966	2	111.92962	8.38	N/Cyto
*Csa2G433370*	3579	1193	4	136.25234	6.98	N
*Csa2G435460*	2721	907	2	104.05611	7.23	N
*Csa3G172400*	3444	1148	1	131.34413	7.38	N/Cyto
*Csa3G684170*	3267	1089	1	125.61899	6.10	Cyto/N
*Csa3G814390*	2409	803	1	93.33195	9.15	N
*Csa3G814400*	2466	822	1	94.42609	7.10	N
*Csa3G815400*	2676	892	1	103.11633	6.43	N
*Csa3G822360*	3159	1053	2	120.24954	6.27	N
*Csa4G015840*	3261	1087	1	124.35665	6.84	Cyto/N
*Csa4G015850*	3240	1080	1	123.31024	7.36	N/Cyto
*Csa4G016360*	2874	958	2	109.57171	7.09	N
*Csa4G016430*	2520	840	3	96.12670	6.14	Cyto/N
*Csa4G016460*	2841	947	2	108.23233	7.33	N
*Csa4G638480*	2463	821	5	94.24655	6.58	N
*Csa5G165310*	879	293	1	32.98491	6.80	Cyto
*Csa5G266890*	3141	1047	1	119.3791	6.04	Cyto/N
*Csa5G647550*	3213	1071	6	121.78672	6.70	N
*Csa5G647580*	1143	381	3	44.46930	8.70	N/Cyto
*Csa5G647590*	1884	628	7	71.53999	5.97	N
*Csa6G375730*	2751	917	2	106.78455	8.21	N
*Csa6G490170*	2454	818	5	93.59007	6.05	N
*Csa7G239020*	3621	1207	2	138.19007	7.59	N/Cyto
*Csa7G420890*	1665	555	2	64.71685	7.61	N
*Csa7G425940*	3060	1020	5	116.40068	6.86	N

N represents nucleus; Cyto represents cytoplasm.

## Data Availability

Not applicable.

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
