# Peer review of "Genome-Wide Identification and Characterization of the CC-NBS-LRR Gene Family in Cucumber (Cucumis sativus L.)"

_ijms, 2022, doi:10.3390/ijms23095048_

Round 1

Reviewer 1 Report

The Ms entitled Genome-wide identification and characterization of the CC-NBS-LRR family genes in cucumber (Cucumis sativus L.) is a very fascinating study, and authors did an excellent job.

But I found a few major issues that must be addressed.

  1. The manuscript is written in a very illegible language. The manuscript must be rewritten in proper English. It is difficult to understand what you want to convey from your research in some paragraphs.
  2. You used qRT-PCR to validate the expression data under biotic stress, tissue development stages, and exogenous hormone treatments. The expression data was also provided under Nacl and low temperature stress, but qRT-PCR is not performed. So, qRT-PCR can be performed under these conditions.
  3. You can also conduct a protein-protein interaction study to better understand the interaction of the NBS-LRR protein.
  4. The family name should not be italicized, correct it in the whole manuscript.
  5. Follow the same trend to write the name of plants, either write scientific names or write normal names
  6. The word whose abbreviation was mentioned once, should not be used again, only use abbreviated form.
  7. Recheck all the names of gene, all should be italicized.
  8. The manuscript contains numerous typographical errors.

Introduction

  1. Line no. 48-49, is not written properly. Please rephrase it.

2.In Line no. 50-51, 62-63, plants only resist the invasion of bacteria or all the pathogens.

  1. In Line no. 74, please give the full form of all the motifs.
  2. In Line no. 88, please give the full form of BOI proteins
  3. From Line no. 100-118, this complete paragraph should not be included in the introduction, it should be added in the discussion part.
  4. Line no. 119-121, should be rephrased.

Results

  1. In Line no. 131 you have mentioned eight genes on chromosome 2, but figure is showing 10 genes on chromosome 2.
  2. In Line no. 135, exited should be replaced with existed.
  3. In heading 2.1, you have not discussed the results of molecular weight, and in table 1, write the molecular weight in kilodalton not in Dalton as you have mentioned in the heading.
  4. In the table 1 footnotes, use represent instead of present.
  5. Line no. 190-193 are not correctly written.
  6. In Line no. 268, you have mentioned the up-regulation of gene Csa3G172400, but heat maps show very less up-regulation as compared to gene Csa2G075440.
  7. In Line no. 319-321, you have mentioned expression level of Csa4G016460 in S1003 was significantly higher than that in NIL(Pm5.1) at 12, 24 and 48 hpi, but graph shows the higher expression of NIL (Pm5.1) at initial hour.
  8. In Line no. 325-326, you have mentioned there is no significant difference between two lines except 24 hpi, but graph shows significant difference at 12 hpi also.
  9. In various places you have write the discussion part with results. That should be avoided and mentioned all these lines in the discussion only.

Materials and Methods

1.There is a repetition of Line no. 527-528.

  1. Paragraph Analysis of gene characteristics, genomic distribution, and cis-acting elements in promotors should be rewrite.
  2. Heading 4.5 should be clearly defined.

Author Response

Responses to editor’s and reviewers’ comments

Thank you so much for your comments on our manuscript entitled “Genome-wide identification and characterization of the CC-NBS-LRR family genes in cucumber (Cucumis sativus L.)”. These comments are extremely important for us to improve the quality of this manuscript. We have revised the original manuscript based on the reviewers’ comments. Our detailed responses are given below.

Responses to Reviewer 1

Dear Reviewer 1,

Thank you so much for your comments on our manuscript entitled “Genome-wide identification and characterization of the CC-NBS-LRR family genes in cucumber (Cucumis sativus L.)”. We really appreciate your comments.

  1. The manuscript is written in a very illegible language. The manuscript must be rewritten in proper English. It is difficult to understand what you want to convey from your research in some paragraphs.

Response: Thanks for your valuable comment. Some paragraphs have been rewritten according to the comment. And the manuscript has been carefully revised by a native English speaker to improve the grammar and readability. If necessary, the manuscript could be checked by the more specialized experts.

  1. You used qRT-PCR to validate the expression data under biotic stress, tissue development stages, and exogenous hormone treatments. The expression data was also provided under Nacl and low temperature stress, but qRT-PCR is not performed. So, qRT-PCR can be performed under these conditions.

Response: Thanks for your valuable comment. Studies have reported that CNL family genes were mainly functioned in disease resistance and were induced by pathogens, such as the case in cabbage (Liu et al., 2019; Liu et al., 2021). In this study, we firstly analyzed the expression of the CNL genes using the published data of previous studies to make heat maps. The main purpose of this study is to analyze the expression pattern responding to biotic stress in cucumber, and the hormones SA, JA and ethylene always functioned under biotic stress. Therefore, we used qRT-PCR to analyze the expression data under biotic stress, for example powdery mildew inoculation, and under treatments of hormones of SA, JA and ethylene. However, the CNL genes might be responding to abiotic stress in cucumber, and the expression data under Nacl and low temperature stress will be in future research.

Liu, Z.; Xie, J.; Wang, H.; Zhong, X.; Li, H.; Yu, J.; Kang, J., Identification and expression profiling analysis of NBS-LRR genes involved in Fusarium oxysporum f.sp. conglutinans resistance in cabbage. 3 Biotech 2019, 9, (5), 202.

Liu, Y.; Li, D.; Yang, N.; Zhu, X.; Han, K.; Gu, R.; Bai, J.; Wang, A.; Zhang, Y., Genome-Wide Identification and Analysis of CC-NBS-LRR Family in Response to Downy Mildew and Black Rot in Chinese Cabbage. Int J Mol Sci 2021, 22, (8).

  1. You can also conduct a protein-protein interaction study to better understand the interaction of the NBS-LRR protein.

Response: Thanks for your valuable comment. Studies have reported that CNL family genes were mainly functioned in disease resistance, and the main content of this study was the bioinformatics analysis and the expression of CNL genes. Powdery mildew is a common and serious disease of cucumber worldwide. We have performed research on powdery mildew resistance in previous studies (Nie et al., 2015a, b; Nie et al., 2021). Therefore, we are attempt to conduct the protein-protein interaction study in next study and uncover the mechanism of the NBS-LRR protein responding to powdery mildew.

Nie J, He H, Peng J, Yang X, Bie B, Zhao J, Wang Y, Si L, Pan J, Cai R. Identification and fine mapping of pm5.1: a recessive gene for powdery mildew resistance in cucumber (Cucumis sativus L.). Molecular Breeding, 2015a, 35(1): 1-13.

Nie J, Wang Y, He H, Guo C, Zhu W, Pan J, Li D, Lian H, Pan J, Cai R. Loss-of-Function Mutations in CsMLO1 Confer Durable Powdery Mildew Resistance in Cucumber (Cucumis sativus L.). Frontiers in Plant Science, 2015b, 6:1155.

Nie J, Wang H, Zhang W, et al. Characterization of lncRNAs and mRNAs involved in powdery mildew resistance in cucumber. Phytopathology, 2021, 111(9): 1613-1624.

  1. The family name should not be italicized, correct it in the whole manuscript.

Response: Thanks for your valuable comment. We have corrected all the related issues in the whole manuscript. For example, Line no.19, 26, 46, 98,123, 124, 127, 129, 132.

  1. Follow the same trend to write the name of plants, either write scientific names or write normal names

Response: Thanks for your valuable comment. We have corrected this issue in the whole manuscript. We have uniformly written all plant names to normal names in the manuscript. For example, Line no.95, 244-247, 250-267.

  1. The word whose abbreviation was mentioned once, should not be used again, only use abbreviated form.

Response: Thanks for your valuable comment. We have corrected that in the whole manuscript. For example, Line no.43, 47, 64, 78, 86.

  1. Recheck all the names of gene, all should be italicized.

Response: Thanks for your carefulness. We have corrected all the related issues in the whole manuscript.  All the names of gene in the manuscript were italicized, and all the names of proteins were normal form.

  1. The manuscript contains numerous typographical errors.

Response: Thanks for your valuable comment. We carefully checked in the manuscript and corrected it by a native English speaker to improve the grammar and readability. If necessary, the manuscript could be checked by the more specialized experts.

Introduction

  1. Line no. 48-49, is not written properly. Please rephrase it.

Response: Thanks for your valuable comment. We have rephrased this sentence in the manuscript. Line no.48-49 “Plants are constantly confronted by various pathogens and external adverse environment cues at different developmental stages.”

  1. In Line no. 50-51, 62-63, plants only resist the invasion of bacteria or all the pathogens.

Response: Thanks for your valuable comment. The meaning is that CNL genes could resist all the pathogens. We have corrected that in the manuscript. Line no.49-50“Plants have developed two layers of defensive mechanisms to resist the invasion of pathogens during this long-term evolutionary process”, Line no.61-62“The disease resistance gene (R gene) is a dominant resistant gene in plants and it can specifically detect pathogens to trigger resistance to disease”

  1. In Line no. 74, please give the full form of all the motifs.

Response: Thanks for your carefulness. We have added the full form of all the motifs. Line no.72-76. “P-LOOP (Phosphate-binding loop), GLPL (also called kinase 3), RNBS-A (Resistance Nucleotide Binding Site-A), RNBS-B (Resistance Nucleotide Binding Site-B), RNBS-C (Resistance Nucleotide Binding Site-C), RNBS-D (Resistance Nucleotide Binding Site-D), kinase 2 and MHDV (Met-His-Asp-Val)”

  1. In Line no. 88, please give the full form of BOI proteins

Response: Thanks for your valuable comment. We have added this full form of BOI protein in the manuscript. Line no.87-88“BOTRYTIS SUSCEPTIBLE1 INTERACTOR(BOI)”

  1. From Line no. 100-118, this complete paragraph should not be included in the introduction, it should be added in the discussion part.

Response: Thanks for your valuable comment. Based on your advice, we have added the complete paragraph (Line no.491-509) in the discussion part in the manuscript.

  1. Line no. 119-121, should be rephrased.

Response: Thanks for your valuable comment. We have rephrased this section in the manuscript. Line no.99-101“In this study, 33 CNL genes were analyzed in cucumber through bioinformatic method. The distribution, gene cluster, characteristics of the CNL family members were analyzed. The Cis-elements of the promoters and conserved motifs of encoded proteins of these CNLs were predicted.”

Results

  1. In Line no. 131 you have mentioned eight genes on chromosome 2, but figure is showing 10 genes on chromosome 2.

Response: Thanks for your carefulness. We have corrected this mistake in the manuscript. Line no.118-119 “The majority of CNL genes in cucumber were located on chromosome 2 with 10 CNL genes”

  1. In Line no. 135, exited should be replaced with existed.

Response: Thanks for your carefulness. We have corrected that in the manuscript. Line no.124-125“there was no gene cluster existed on chromosomes 1 and 6”

  1. In heading 2.1, you have not discussed the results of molecular weight, and in table 1, write the molecular weight in kilodalton not in Dalton as you have mentioned in the heading.

Response: Thanks for your carefulness. Firstly, we have added the discussion about the results of molecular weight in the manuscript (Line no.133-135). Secondly, we have rephrased the molecular weight of CNL genes in kilodalton instead of Dalton (Line no.144, Table1).

  1. In the table 1 footnotes, use represent instead of present.

Response: Thanks for your valuable comment. We have corrected the presentation in the manuscript. Line no.145“N represent Nucleus; Cyto represent Cytoplasm.”

  1. Line no. 190-193 are not correctly written.

Response: Thanks for your valuable comment. We have rewritten this sentence in the manuscript. Line no.170-174 “We found that most of the amino acid sequences were arranged by order of ‘motif13-motif9-motif7-motif16-motif14-motif20-motif11-motif2-motif1-motif10-motif3-motif5-motif8-motif6-motif19-motif4-motif17-motif15-motif18-motif12’.”

  1. In Line no. 268, you have mentioned the up-regulation of gene Csa3G172400, but heat maps show very less up-regulation as compared to gene Csa2G075440.

Response: Thanks for your valuable comment. We checked the expression of Csa2G075440 in the supplementary table 6, and the degree of expression variation of Csa2G075440 was small. Therefore, we use Csa5G647590 in place of Csa2G075440. Line no.247-248 “such as Csa5G647590 and Csa3G172400

  1. In Line no. 319-321, you have mentioned expression level of Csa4G016460 in S1003 was significantly higher than that in NIL(Pm5.1) at 12, 24 and 48 hpi, but graph shows the higher expression of NIL (Pm5.1) at initial hour.

Response: Thanks for your valuable comment. We checked the data and added the significantly difference in the figure and rephrased in the manuscript. Line no.294-296 “The expression level of Csa4G016460, Csa3G172400 and Csa4G638480 in S1003 were significantly lower than that in NIL(Pm5.1) at 0 hpi.”

  1. In Line no. 325-326, you have mentioned there is no significant difference between two lines except 24 hpi, but graph shows significant difference at 12 hpi also.

Response: Thanks for your valuable comment. We checked the data and rephrased this section in the manuscript. Line no.306-307 “Except for 0 and 24 hpi, there was significantly difference of the expression level of Csa2G435460 between S1003 and NIL(Pm5.1).”

  1. In various places you have write the discussion part with results. That should be avoided and mentioned all these lines in the discussion only.

Response: Thanks for your valuable comment. We have checked all results in the manuscript, and removed the related presentation to the discussion part.

Materials and Methods

1.There is a repetition of Line no. 527-528.

Response: Thanks for your valuable comment. We have corrected that in the manuscript. Line no.534-536.

  1. Paragraph Analysis of gene characteristics, genomic distribution, and cis-acting elements in promotors should be rewrite.

Response: Thanks for your valuable comment. We have rewritten this paragraph in the manuscript. Line no.541-555.

  1. Heading 4.5 should be clearly defined.

Response: Thanks for your valuable comment.  We have rewritten this heading and added another heading to clearly defined the treatment. Line no.537-591.

Reviewer 2 Report

The presented article «Genome-wide identification and characterization of the CC-NBS-LRR family genes in cucumber (Cucumis sativus L» is devoted to important and topical problems of molecular biology, which have a pronounced applied value for plant biotechnology. This article is of great interest to the world scientific community. Overall, the manuscript is well organized. However, unfortunately, there are some remarks, that should be taken into account.

  • Add of the author-classifier to the genus and species names plants (lines 93, 212, 225, 229-246, 441 etc.). Additionally, for the first time in the text, the generic name should be presented in full (for example, Solanum), while further in the text it should be abbreviated (for example, S.). The authors use the old tomato nomenclature (Lycopersicum esculentum ). Correction to the modern Latin name is required (Solanum lycopersicum L.) (lines 231, 240, fig. 4 etc)
  • Line 57. ETI (effector-triggered immunity) → effector-triggered immunity (ETI);
  • Lines 62, 63 etc. R gene (italic)
  • The introduction could use some citations in specific locations (such as lines 66);
  • Lines 93, cotton (not italic) or full Latin name;
  • Some lines are not appropriate in the Results section (157-157: « Cis-elements play an important role in various life processes, such as plant growth and development, stress response, hormones response, and signal transduction» and 175-181: « Among them, 16 gene members contained an element named “W-box”, which is the DNA binding site of WRKY protein induced by salicylic acid[40]. Research suggested that it 177 could positively regulate RPP8, a downy mildew resistance gene, in Arabidopsis[41]. The results suggested that the CNL genes might be implicated in the defense against certain fungal diseases, and the CNL genes might also play a role in defense against various dis eases in cucumber.» They should be moved to the Introduction or Discussion section.
  • Line 198 Csa5G647580 and Csa5G165310 (italic);
  • 7-11. Error bars represent the standard error based on the mean of three biological replicates. A number of questions arise. What do the authors mean by biological replicates? How many samples in each replicate (n=?)? How many independent plants were used for each treatment? All these important data must also be accompanied in the Materials and Methods section.
  • There is no conclusion section.

Author Response

Responses to editor’s and reviewers’ comments

Thank you so much for your comments on our manuscript entitled “Genome-wide identification and characterization of the CC-NBS-LRR family genes in cucumber (Cucumis sativus L.)”. These comments are extremely important for us to improve the quality of this manuscript. We have revised the original manuscript based on the reviewers’ comments. Our detailed responses are given below.

Responses to Reviewer 2

Dear Reviewer 2,

Thank you so much for your comments on our manuscript entitled “Genome-wide identification and characterization of the CC-NBS-LRR family genes in cucumber (Cucumis sativus L.)”. We really appreciate your comments.

  1. Add of the author-classifier to the genus and species names plants (lines 93, 212, 225, 229-246, 441 etc.). Additionally, for the first time in the text, the generic name should be presented in full (for example, Solanum), while further in the text it should be abbreviated (for example, S.). The authors use the old tomato nomenclature (Lycopersicum esculentum ). Correction to the modern Latin name is required (Solanum lycopersicum ) (lines 231, 240, fig. 4 etc)

Response: Thanks for your valuable comments. We have rephrased all plant names by using normal name. Line no. 92, 195, 198, 211-225, 231-232, 418. The full name of the generic name was added. We also corrected tomato nomenclature in the Figure 4 and text. Line no.231.

  1. Line 57. ETI (effector-triggered immunity) → effector-triggered immunity (ETI);

Response: Thanks for your valuable comments. We have corrected that in the manuscript. Line no.57.

  1. Lines 62, 63 etc. R gene (italic)

Response: Thanks for pointing out this problem. We have corrected the format in the manuscript. Line no.61,62. All similar issues were checked and corrected in the manuscript.

  1. The introduction could use some citations in specific locations (such as lines 66);

Response: Thanks for your valuable comments. We have added some citations in introduction. Line no.65, 80.

  1. Lines 93, cotton (not italic) or full Latin name;

Response: Thanks for your valuable comments. We have rephrased cotton in the manuscript. Line no.93. All plant names were rewritten by using normal names in the manuscript. Line no.93, 95,

  1. Some lines are not appropriate in the Results section (157-157: « Cis-elements play an important role in various life processes, such as plant growth and development, stress response, hormones response, and signal transduction» and 175-181: « Among them, 16 gene members contained an element named “W-box”, which is the DNA binding site of WRKY protein induced by salicylic acid[40]. Research suggested that it 177 could positively regulate RPP8, a downy mildew resistance gene, in Arabidopsis[41]. The results suggested that the CNL genes might be implicated in the defense against certain fungal diseases, and the CNL genes might also play a role in defense against various dis eases in cucumber.» They should be moved to the Introduction or Discussion section.

Response: Thanks for your valuable comments. We have moved these sections to the discussion part in the manuscript. Line no.423-424, 427-432. Meanwhile, we have checked and corrected other similar presentation.

  1. Line 198 Csa5G647580 and Csa5G165310 (italic);

Response: Thanks for pointing out this problem. We have rewritten the italic form of Csa5G647580 gene and Csa5G165310 gene in the manuscript. Lin no.181.

  1. 7-11. Error bars represent the standard error based on the mean of three biological replicates. A number of questions arise. What do the authors mean by biological replicates? How many samples in each replicate (n=?)? How many independent plants were used for each treatment? All these important data must also be accompanied in the Materials and Methods section.

Response: Thanks for pointing out this problem. We have added the information of the number of samples in each replicate in the Material and Method section. Line no. 578-579, 587-588, 592-593.

  1. There is no conclusion section.

Response: Thanks for your valuable comments. We added conclusion section in the manuscript. Line no.611-630.

Round 2

Reviewer 1 Report

Revised manuscript has been significantly improved.